# STABLE OPPONENT SHAPING
# IN DIFFERENTIABLE GAMES

**Alistair Letcher**[1]  **Jakob Foerster**[1]  **David Balduzzi**[2]  **Tim Rocktäschel**[3]  **Shimon Whiteson**[1]

[1]University of Oxford   [2]DeepMind   [3]University College London

## ABSTRACT

A growing number of learning methods are actually *differentiable games* whose players optimise multiple, interdependent objectives in parallel – from GANs and intrinsic curiosity to multi-agent RL. Opponent shaping is a powerful approach to improve learning dynamics in these games, accounting for player influence on others' updates. Learning with Opponent-Learning Awareness (LOLA) is a recent algorithm that exploits this response and leads to cooperation in settings like the Iterated Prisoner's Dilemma. Although experimentally successful, we show that LOLA agents can exhibit 'arrogant' behaviour directly at odds with convergence. In fact, remarkably few algorithms have theoretical guarantees applying across all ($n$-player, non-convex) games. In this paper we present Stable Opponent Shaping (SOS), a new method that interpolates between LOLA and a stable variant named LookAhead. We prove that LookAhead converges locally to equilibria and avoids strict saddles in *all differentiable games*. SOS inherits these essential guarantees, while also shaping the learning of opponents and consistently either matching or outperforming LOLA experimentally.

## 1 INTRODUCTION

**Problem Setting.**  While machine learning has traditionally focused on optimising single objectives, generative adversarial nets (GANs) (Goodfellow et al., 2014) have showcased the potential of architectures dealing with *multiple* interacting goals. They have since then proliferated substantially, including intrinsic curiosity (Pathak et al., 2017), imaginative agents (Racanière et al., 2017), synthetic gradients (Jaderberg et al., 2017), hierarchical reinforcement learning (RL) (Wayne & Abbott, 2014; Vezhnevets et al., 2017) and multi-agent RL in general (Busoniu et al., 2008).

These can effectively be viewed as *differentiable games* played by cooperating and competing *agents* – which may simply be different internal components of a single system, like the generator and discriminator in GANs. The difficulty is that each loss depends on *all* parameters, including those of other agents. While gradient descent on single functions has been widely successful, converging to local minima under rather mild conditions (Lee et al., 2017), its simultaneous generalisation can fail even in simple two-player, two-parameter zero-sum games. No algorithm has yet been shown to converge, even locally, in all differentiable games.

**Related Work.**  Convergence has widely been studied in *convex* $n$-player games, see especially Rosen (1965); Facchinei & Kanzow (2007). However, the recent success of *non-convex* games exemplified by GANs calls for a better understanding of this general class where comparatively little is known. Mertikopoulos & Zhou (2018) recently prove local convergence of no-regret learning to *variationally stable* equilibria, though under a number of regularity assumptions.

Conversely, a number of algorithms have been successful in the non-convex setting for *restricted classes* of games. These include policy prediction in two-player two-action bimatrix games (Zhang & Lesser, 2010); WoLF in two-player two-action games (Bowling & Veloso, 2001); AWESOME in repeated games (Conitzer & Sandholm, 2007); Optimistic Mirror Descent in two-player bilinear zero-sum games (Daskalakis et al., 2018) and Consensus Optimisation (CO) in two-player zero-sum games (Mescheder et al., 2017). An important body of work including Heusel et al. (2017); Nagarajan & Kolter (2017) has also appeared for the specific case of GANs.

Working towards bridging this gap, some of the authors recently proposed Symplectic Gradient Adjustment (SGA), see Balduzzi et al. (2018). This algorithm is provably 'attracted' to stable fixed points while 'repelled' from unstable ones in *all* differentiable games ($n$-player, non-convex). Nonetheless, these results are weaker than strict convergence guarantees. Moreover, SGA agents may act against their own self-interest by prioritising stability over individual loss. SGA was also discovered independently by Gemp & Mahadevan (2018), drawing on variational inequalities.

In a different direction, Learning with Opponent-Learning Awareness (LOLA) (Foerster et al., 2018) modifies the learning objective by predicting and differentiating through opponent learning steps. This is intuitively appealing and experimentally successful, encouraging cooperation in settings like the Iterated Prisoner's Dilemma (IPD) where more stable algorithms like SGA defect. However, LOLA has no guarantees of converging or even preserving fixed points of the game.

**Contribution.** We begin by constructing the first explicit *tandem* game where LOLA agents adopt 'arrogant' behaviour and converge to non-fixed points. We pinpoint the cause of failure and show that a natural variant named LookAhead (LA), discovered before LOLA by Zhang & Lesser (2010), successfully preserves fixed points. We then prove that LookAhead locally converges and avoids strict saddles in all differentiable games, filling a theoretical gap in multi-agent learning. This is enabled through a unified approach based on fixed-point iterations and dynamical systems. These techniques apply equally well to algorithms like CO and SGA, though this is not our present focus.

While LookAhead is theoretically robust, the shaping component endowing LOLA with a capacity to exploit opponent dynamics is lost. We solve this dilemma with an algorithm named Stable Opponent Shaping (SOS), trading between stability and exploitation by interpolating between LookAhead and LOLA. Using an intuitive and theoretically grounded criterion for this interpolation parameter, SOS inherits both strong convergence guarantees from LA and opponent shaping from LOLA.

On the experimental side, we show that SOS plays tit-for-tat in the IPD on par with LOLA, while all other methods mostly defect. We display the practical consequences of our theoretical guarantees in the tandem game, where SOS always outperforms LOLA. Finally we implement a more involved GAN setup, testing for mode collapse and mode hopping when learning Gaussian mixture distributions. SOS successfully spreads mass across all Gaussians, at least matching dedicated algorithms like CO, while LA is significantly slower and simultaneous gradient descent fails entirely.

## 2 BACKGROUND

### 2.1 DIFFERENTIABLE GAMES

We frame the problem of multi-agent learning as a game. Adapted from Balduzzi et al. (2018), the following definition insists only on differentiability for gradient-based methods to apply. This concept is strictly more general than stochastic games, whose parameters are usually restricted to action-state transition probabilities or functional approximations thereof.

**Definition 1.** A *differentiable game* is a set of $n$ players with parameters $\theta = (\theta^1, \ldots, \theta^n) \in \mathbb{R}^d$ and twice continuously differentiable losses $L^i : \mathbb{R}^d \to \mathbb{R}$, where $\theta^i \in \mathbb{R}^{d_i}$ for each $i$ and $\sum_i d_i = d$.

Crucially, note that each loss is a function of *all* parameters. From the viewpoint of player $i$, parameters can be written as $\theta = (\theta^i, \theta^{-i})$ where $\theta^{-i}$ contains all other players' parameters. We do not make the common assumption that each $L^i$ is convex as a function of $\theta^i$ alone, for any fixed opponent parameters $\theta^{-i}$, nor do we restrict $\theta$ to the probability simplex – though this restriction can be recovered via projection or sigmoid functions $\sigma : \mathbb{R} \to [0, 1]$. If $n = 1$, the 'game' is simply to minimise a given loss function. In this case one can reach *local minima* by (possibly stochastic) gradient descent (GD). For arbitrary $n$, the standard solution concept is that of *Nash equilibria*.

**Definition 2.** A point $\bar{\theta} \in \mathbb{R}^d$ is a (local) Nash equilibrium if for each $i$, there are neighbourhoods $U_i$ of $\bar{\theta}^i$ such that $L^i(\theta^i, \bar{\theta}^{-i}) \geq L^i(\bar{\theta})$ for all $\theta^i \in U_i$. In other words, each player's strategy is a local *best response* to current opponent strategies.

We write $\nabla_i L^k = \nabla_{\theta^i} L^k$ and $\nabla_{ij} L^k = \nabla_{\theta^j} \nabla_{\theta^i} L^k$ for any $i, j, k$. Define the *simultaneous gradient* of the game as the concatenation of each player's gradient,

$$\xi = \left( \nabla_1 L^1, \ldots, \nabla_n L^n \right)^\mathsf{T} \in \mathbb{R}^d.$$

The $i$th component of $\xi$ is the direction of greatest increase in $L^i$ with respect to $\theta^i$. If each agent minimises their loss independently from others, they perform GD on their component $\nabla_i L^i$ with learning rate $\alpha_i$. Hence, the parameter update for all agents is given by $\theta \leftarrow \theta - \alpha \odot \xi$, where $\alpha = (\alpha_1, \ldots, \alpha_n)^\intercal$ and $\odot$ is element-wise multiplication. This is also called naive learning (NL), reducing to $\theta \leftarrow \theta - \alpha\xi$ if agents have the same learning rate. This is assumed for notational simplicity, though irrelevant to our results. The following example shows that NL can fail to converge.

**Example 1.** Consider $L^{1/2} = \pm xy$, where players control the $x$ and $y$ parameters respectively. The origin is a (global and unique) Nash equilibrium. The simultaneous gradient is $\xi = (y, -x)$ and cycles around the origin. Explicitly, a gradient step from $(x, y)$ yields

$$(x, y) \leftarrow (x, y) - \alpha(y, -x) = (x - \alpha y, y + \alpha x)$$

which has distance from the origin $(1 + \alpha^2)(x^2 + y^2) > (x^2 + y^2)$ for any $\alpha > 0$ and $(x, y) \neq 0$. It follows that agents diverge away from the origin for any $\alpha > 0$. The cause of failure is that $\xi$ is not the gradient of a single function, implying that each agent's loss is inherently dependent on others. This results in a contradiction between the non-stationarity of each agent, and the optimisation of each loss independently from others. Failure of convergence in this simple two-player zero-sum game shows that gradient descent does not generalise well to differentiable games. We consider an alternative solution concept to Nash equilibria before introducing LOLA.

## 2.2 STABLE FIXED POINTS

Consider the game given by $L^1 = L^2 = xy$ where players control the $x$ and $y$ parameters respectively. The optimal solution is $(x, y) \to \pm(\infty, -\infty)$, since then $L^1 = L^2 \to -\infty$. However the origin is a *global Nash equilibrium*, while also a *saddle point* of $xy$. It is highly undesirable to converge to the origin in this game, since infinitely better losses can be reached in the anti-diagonal direction. In this light, Nash equilibria cannot be the right solution concept to aim for in multi-agent learning. To define stable fixed points, first introduce the 'Hessian' of the game as the block matrix

$$H = \nabla\xi = \begin{pmatrix} \nabla_{11}L^1 & \cdots & \nabla_{1n}L^1 \\ \vdots & \ddots & \vdots \\ \nabla_{n1}L^n & \cdots & \nabla_{nn}L^n \end{pmatrix} \in \mathbb{R}^{d \times d}.$$

This can equivalently be viewed as the Jacobian of the vector field $\xi$. Importantly, note that $H$ is not symmetric in general unless $n = 1$, in which case we recover the usual Hessian $H = \nabla^2 L$.

**Definition 3.** A point $\bar{\theta}$ is a *fixed point* if $\xi(\bar{\theta}) = 0$. It is *stable* if $H(\bar{\theta}) \succeq 0$, *unstable* if $H(\bar{\theta}) \prec 0$ and a *strict saddle* if $H(\bar{\theta})$ has an eigenvalue with negative real part.

The name 'fixed point' is coherent with GD, since $\xi(\bar{\theta}) = 0$ implies a fixed update $\bar{\theta} \leftarrow \bar{\theta} - \alpha\xi(\bar{\theta}) = \bar{\theta}$. Though Nash equilibria were shown to be inadequate above, it is not obvious that stable fixed points (SFPs) are a better solution concept. In Appendix A we provide intuition for why SFPs are both closer to local minima in the context of multi-loss optimisation, and more tractable for convergence proofs. Moreover, this definition is an improved variant on that in Balduzzi et al. (2018), assuming positive semi-definiteness only at $\bar{\theta}$ instead of holding in a neighbourhood. This makes the class of SFPs as large as possible, while sufficient for all our theoretical results.

Assuming invertibility of $H(\bar{\theta})$ at SFPs is crucial to all convergence results in this paper. The same assumption is present in related work including Mescheder et al. (2017), and cannot be avoided. Even for single losses, a fixed point with singular Hessian can be a local minimum, maximum, or saddle point. Invertibility is thus necessary to ensure that SFPs really are 'local minima'. This is omitted from now on. Finally note that unstable fixed points are a *subset* of strict saddles, making Theorem 6 both stronger and more general than results for SGA by Balduzzi et al. (2018).

## 2.3 LEARNING WITH OPPONENT-LEARNING AWARENESS (LOLA)

Accounting for nonstationarity, Learning with Opponent-Learning Awareness (LOLA) modifies the learning objective by predicting and differentiating through opponent learning steps (Foerster et al., 2018). For simplicity, if $n = 2$ then agent 1 optimises $L^1(\theta^1, \theta^2 + \Delta\theta^2)$ with respect to $\theta^1$, where

$\Delta\theta^2$ is the predicted learning step for agent 2. Foerster et al. (2018) assume that opponents are naive learners, namely $\Delta\theta^2 = -\alpha_2 \nabla_2 L^2$. After first-order Taylor expansion, the loss is approximately given by $L^1 + \nabla_2 L^1 \cdot \Delta\theta^2$. By minimising this quantity, agent 1 learns parameters that align the opponent learning step $\Delta\theta^2$ with the direction of greatest decrease in $L^1$, exploiting opponent dynamics to further reduce one's losses. Differentiating with respect to $\theta^1$, the adjustment is

$$\nabla_1 L^1 + \left(\nabla_{21} L^1\right)^\mathsf{T} \Delta\theta^2 + \left(\nabla_1 \Delta\theta^2\right)^\mathsf{T} \nabla_2 L^1 \,.$$

By explicitly differentiating through $\Delta\theta^2$ in the rightmost term, LOLA agents actively shape opponent learning. This has proven effective in reaching cooperative equilibria in multi-agent learning, finding success in a number of games including tit-for-tat in the IPD. The middle term above was originally dropped by the authors because "LOLA focuses on this shaping of the learning direction of the opponent". We choose *not* to eliminate this term, as also inherent in LOLA-DiCE (Foerster et al., 2018). Preserving both terms will in fact be key to developing *stable* opponent shaping.

First we formulate $n$-player LOLA in vectorial form. Let $H_d$ and $H_o$ be the matrices of diagonal and anti-diagonal blocks of $H$, so that $H = H_d + H_o$. Also define $L = (L^1, \ldots, L^n)$ and the operator $\mathrm{diag} : \mathbb{R}^{d \times n} \to \mathbb{R}^d$ constructing a vector from the block matrix diagonal, namely $\mathrm{diag}(M)_i = M_{ii}$.

**Proposition 1** (Appendix B). Writing $\chi = \mathrm{diag}(H_o^\mathsf{T} \nabla L)$, the LOLA gradient adjustment is

$$\textsc{lola} = (I - \alpha H_o)\xi - \alpha\chi \,.$$

While experimentally successful, LOLA fails to preserve fixed points $\bar\theta$ of the game since

$$(I - \alpha H_o)\xi(\bar\theta) - \alpha\chi(\bar\theta) = -\alpha\chi(\bar\theta) \neq 0$$

in general. Even if $\bar\theta$ is a Nash equilibrium, the update $\bar\theta \leftarrow \bar\theta - \alpha\textsc{lola} \neq \bar\theta$ can push them away despite parameters being optimal. This may worsen the losses for all agents, as in the game below.

**Example 2** (Tandem). Imagine a tandem controlled by agents facing opposite directions, who feed $x$ and $y$ force into their pedals respectively. Negative numbers correspond to pedalling backwards.

Moving coherently requires $x \approx -y$, embodied by a quadratic loss $(x+y)^2$. However it is easier for agents to pedal forwards, translated by linear losses $-2x$ and $-2y$. The game is thus given by $L^1(x, y) = (x + y)^2 - 2x$ and $L^2(x, y) = (x + y)^2 - 2y$. These sub-goals are incompatible, so agents cannot simply accelerate forwards. The SFPs are given by $\{x + y = 1\}$. Computing $\chi(x, 1 - x) = (4, 4) \neq 0$, none of these are preserved by LOLA. Instead, we show in Appendix C that LOLA can only converge to sub-optimal scenarios with worse losses for *both* agents, for *any* $\alpha$.

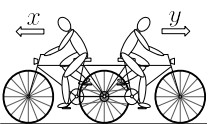

Figure 1: Illustration of the tandem game.

Intuitively, the root of failure is that LOLA agents try to shape opponent learning and enforce compliance by accelerating forwards, assuming a dynamic response from their opponent. The other agent does the same, so they become 'arrogant' and suffer by pushing strongly in opposite directions.

## 3 METHOD

### 3.1 LOOKAHEAD

The shaping term $\chi$ prevents LOLA from preserving fixed points. Consider removing this component entirely, giving $(I - \alpha H_o)\xi$. This variant preserves fixed points, but what does it mean from the perspective of each agent? Note that LOLA optimises $L^1(\theta^1, \theta^2 + \Delta\theta^2)$ with respect to $\theta^1$, while $\Delta\theta^2$ is a function of $\theta^1$. In other words, we assume that our opponent's learning step depends on our *current* optimisation with respect to $\theta^1$. This is inaccurate, since opponents cannot see our updated parameters until the *next* step. Instead, assume we optimise $L^1(\theta^1, \hat\theta^2 + \Delta\theta^2(\hat\theta^1, \hat\theta^2))$ where $\hat\theta^1, \hat\theta^2$ are the *current* parameters. After Taylor expansion, the gradient with respect to $\theta^1$ is given by

$$\nabla_1 L^1 + \left(\nabla_{21} L^1\right)^\mathsf{T} \Delta\theta^2$$

since $\Delta\theta^2(\hat\theta^1, \hat\theta^2)$ does not depend on $\theta^1$. In vectorial form, we recover the variant $(I - \alpha H_o)\xi$ since the shaping term corresponds precisely to differentiating through $\Delta\theta^2$. We name this LookAhead, which was discovered *before* LOLA by Zhang & Lesser (2010) though not explicitly named. Using

the stop-gradient operator $\perp^1$, this can be reformulated as optimising $L^1(\theta^1, \theta^2 + \perp \Delta\theta^2)$ where $\perp$ prevents gradient flowing from $\Delta\theta^2$ upon differentiation.

The main result of Zhang & Lesser (2010) is that LookAhead converges to Nash equilibria in the small class of two-player, two-action bimatrix games. We will prove local convergence to SFP and non-convergence to strict saddles in *all differentiable games*. On the other hand, by discarding the problematic shaping term, we also eliminated LOLA's capacity to exploit opponent dynamics and encourage cooperation. This will be witnessed in the IPD, where LookAhead agents mostly defect.

### 3.2 STABLE OPPONENT SHAPING (SOS)

We propose Stable Opponent Shaping (SOS), an algorithm preserving both advantages at once. Define the *partial stop-gradient* operator $\perp^p := p\perp + (1-p)I$, where $I$ is the identity and $p$ stands for *partial*. A $p$-LOLA agent optimises the modified objective

$$L^1(\theta^1, \theta^2 + \perp^{1-p}\Delta\theta^2, \ldots, \theta^n + \perp^{1-p}\Delta\theta^n),$$

collapsing to LookAhead at $p = 0$ and LOLA at $p = 1$. The resulting gradient is given by

$$\xi_p := p\text{-LOLA} = (I - \alpha H_o)\xi - p\alpha\chi$$

with $\xi_0 = \text{LA}$. We obtain an algorithm trading between shaping and stability as a function of $p$. Note however that preservation of fixed points only holds if $p$ is infinitesimal, in which case $p$-LOLA is almost identical to LookAhead – losing the very purpose of interpolation. Instead we propose a two-part criterion for $p$ at each learning step, through which all guarantees descend.

First choose $p$ such that $\xi_p$ points in the same direction as LookAhead. This will not be enough to prove convergence itself, but prevents arrogant behaviour by ensuring convergence *only* to fixed points. Formally, the first criterion is given by $\langle \xi_p, \xi_0 \rangle \geq 0$. If $\langle -\alpha\chi, \xi_0 \rangle \geq 0$ then $\langle \xi_p, \xi_0 \rangle \geq 0$ automatically, so we choose $p = 1$ for maximal shaping. Otherwise choose

$$p = \min\left\{1, \frac{-a\|\xi_0\|^2}{\langle -\alpha\chi, \xi_0 \rangle}\right\}$$

with any hyperparameter $0 < a < 1$. This guarantees a positive inner product

$$\langle \xi_p, \xi_0 \rangle = p\langle -\alpha\chi, \xi_0 \rangle + \|\xi_0\|^2 \geq -a\|\xi_0\|^2 + \|\xi_0\|^2 = \|\xi_0\|^2(1-a) > 0\,.$$

We complement this with a second criterion ensuring local convergence. The idea is to scale $p$ by a function of $\|\xi\|$ if $\|\xi\|$ is small enough, which certainly holds in neighbourhoods of fixed points. Let $0 < b < 1$ be a hyperparameter and take $p = \|\xi\|^2$ if $\|\xi\| < b$, otherwise $p = 1$. Choosing $p_1$ and $p_2$ according to these criteria, the two-part criterion is $p = \min\{p_1, p_2\}$. SOS is obtained by combining $p$-LOLA with this criterion, as summarised in Algorithm 1. Crucially, all theoretical results in the next section are independent from the choice of hyperparameters $a$ and $b$.

---

**Algorithm 1:** Stable Opponent Shaping

---

1 Initialise $\theta$ randomly and fix hyperparameters $a, b \in (0, 1)$.
2 **while** *not done* **do**
3     Compute $\xi_0 = (I - \alpha H_o)\xi$ and $\chi = \text{diag}(H_o^{\mathsf{T}}\nabla L)$ at $\theta$.
4     **if** $\langle -\alpha\chi, \xi_0 \rangle > 0$   **then**   $p_1 = 1$   **else**   $p_1 = \min\left\{1, \frac{-a\|\xi_0\|^2}{\langle -\alpha\chi, \xi_0 \rangle}\right\}$
5     **if** $\|\xi\| < b$   **then**   $p_2 = \|\xi\|^2$   **else**   $p_2 = 1$
6     Let $p = \min\{p_1, p_2\}$, compute $\xi_p = \xi_0 - p\alpha\chi$ and assign $\theta \leftarrow \theta - \alpha\xi_p$.
7 **end**

---

## 4 THEORETICAL RESULTS

Our central theoretical contribution is that LookAhead and SOS converge locally to SFP and avoid strict saddles in *all differentiable games*. Since the learning gradients involve second-order Hessian terms, our results assume thrice continuously differentiable losses (omitted hereafter). Losses which are $C^2$ but not $C^3$ are very degenerate, so this is a mild assumption. Statements made about SOS crucially hold for *any* hyperparameters $a, b \in (0, 1)$. See Appendices D and E for detailed proofs.

---

[1]This operator is implemented in TensorFlow as `stop_gradient` and in PyTorch as `detach`.

## 4.1 LOCAL CONVERGENCE TO STABLE FIXED POINTS

Convergence is proved using Ostrowski's Theorem. This reduces convergence of a gradient adjustment $g$ to positive stability (eigenvalues with positive real part) of $\nabla g$ at stable fixed points.

**Theorem 2.** Let $H \succeq 0$ be invertible with symmetric diagonal blocks. Then there exists $\epsilon > 0$ such that $(I - \alpha H_o)H$ is positive stable for all $0 < \alpha < \epsilon$.

This type of result would usually be proved either by analytical means showing positive definiteness and hence positive stability, or direct eigenvalue analysis. We show in Appendix D that $(I - \alpha H_o)H$ is not necessarily positive definite, while there is no necessary relationship between eigenpairs of $H$ and $H_o$. This makes our theorem all the more interesting and non-trivial. We use a similarity transformation trick to circumvent the dual obstacle, allowing for analysis of positive definiteness with respect to a new inner product. We obtain positive stability by invariance under change of basis.

**Corollary 3.** LookAhead converges locally to stable fixed points for $\alpha > 0$ sufficiently small.

Using the second criterion for $p$, we prove local convergence of SOS in all differentiable games despite the presence of a shaping term (unlike LOLA).

**Theorem 4.** SOS converges locally to stable fixed points for $\alpha > 0$ sufficiently small.

## 4.2 AVOIDING STRICT SADDLES

Using the first criterion for $p$, we prove that SOS only converges to fixed points (unlike LOLA).

**Proposition 5.** If SOS converges to $\bar{\theta}$ and $\alpha > 0$ is small then $\bar{\theta}$ is a fixed point of the game.

Now assume that $\theta$ is initialised randomly (or with arbitrarily small noise), as is standard in ML. Let $F(\theta) = \theta - \alpha \xi_p(\theta)$ be the SOS iteration. Using both the second criterion and the Stable Manifold Theorem from dynamical systems, we can prove that every strict saddle $\bar{\theta}$ has a neighbourhood $U$ such that $\{\theta \in U \mid F^n(\theta) \to \bar{\theta}$ as $n \to \infty\}$ has measure zero for $\alpha > 0$ sufficiently small. Since $\theta$ is initialised randomly, we obtain the following result.

**Theorem 6.** SOS locally avoids strict saddles almost surely, for $\alpha > 0$ sufficiently small.

This also holds for LookAhead, and could be strenghtened to *global* initialisations provided a strong boundedness assumption on $\|H\|_2$. This is trickier for SOS since $p(\theta)$ is not globally continuous. Altogether, our results for LookAhead and the correct criterion for $p$-LOLA lead to some of the strongest theoretical guarantees in multi-agent learning. Furthermore, SOS retains all of LOLA's opponent shaping capacity while LookAhead does not, as shown experimentally in the next section.

## 5 EXPERIMENTS AND DISCUSSION

We evaluate the performance of SOS in three differentiable games. We first showcase opponent shaping and superiority over LA/CO/SGA/NL in the Iterated Prisoner's Dilemma (IPD). This leaves SOS and LOLA, which have differed only in theory up to now. We bridge this gap by showing that SOS always outperforms LOLA in the tandem game, avoiding arrogant behaviour by decaying $p$ while LOLA overshoots. Finally we test SOS on a more involved GAN learning task, with results similar to dedicated methods like Consensus Optimisation.

## 5.1 EXPERIMENTAL SETUP

**IPD:** This game is an infinite sequence of the well-known Prisoner's Dilemma, where the payoff is discounted by a factor $\gamma \in [0, 1)$ at each iteration. Agents are endowed with a memory of actions at the previous state. Hence there are 5 parameters for each agent $i$: the probability $P^i(C \mid state)$ of cooperating at start state $s_0 = \varnothing$ or state $s_t = (a_{t-1}^1, a_{t-1}^2)$ for $t > 0$. One Nash equilibrium is to always defect (DD), with a normalised loss of 2. A better equilibrium with loss 1 is named *tit-for-tat* (TFT), where each player begins by cooperating and then mimicks the opponent's previous action.

We run 300 training episodes for SOS, LA, CO, SGA and NL. The parameters are initialised following a normal distribution around $1/2$ probability of cooperation, with unit variance. We fix $\alpha = 1$ and $\gamma = 0.96$, following Foerster et al. (2018). We choose $a = 0.5$ and $b = 0.1$ for SOS. The first is a robust and arbitrary middle ground, while the latter is intentionally small to avoid poor SFP.

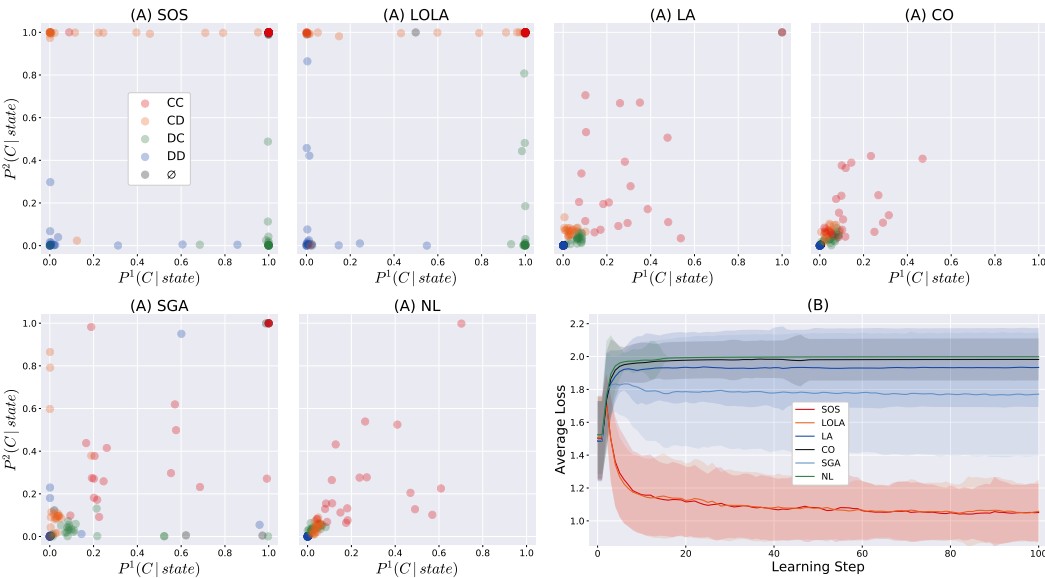

Figure 2: Results in the IPD. **(A)** Probability that agents cooperate, given memory state, at the end of 50 training runs. SOS and LOLA mostly play tit-for-tat, while others mostly defect. **(B)** Average loss at each step, across 300 runs, with shaded deviations. SOS and LOLA outperform all others.

**Tandem:** Though *local* convergence is guaranteed for SOS, it is possible that SOS diverges from poor initialisations. This turns out to be impossible in the tandem game since the Hessian is *globally* positive semi-definite. We show this explicitly by running 300 training episodes for SOS and LOLA. Parameters are initialised following a normal distribution around the origin. We found performance to be robust to hyperparameters $a, b$. Here we fix $a = b = 0.5$ and $\alpha = 0.1$.

**Gaussian mixtures:** We reproduce a setup from Balduzzi et al. (2018). The game is to learn a Gaussian mixture distribution using GANs. Data is sampled from a highly multimodal distribution designed to probe the tendency to collapse onto a subset of modes during training – see ground truth in Appendix F. The generator and discriminator networks each have 6 ReLU layers of 384 neurons, with 2 and 1 output neurons respectively. Learning rates are chosen by grid search at iteration 8k, with $a = 0.5$ and $b = 0.1$ for SOS, following the same reasoning as the IPD.

## 5.2 RESULTS AND DISCUSSION

**IPD:** Results are given in Figure 2. Parameters in part (A) are the end-run probabilities of cooperating for each memory state, encoded in different colours. Only 50 runs are shown for visibility. Losses at each step are displayed in part (B), averaged across 300 episodes with shaded deviations.

SOS and LOLA mostly succeed in playing tit-for-tat, displayed by the accumulation of points in the correct corners of (A) plots. For instance, CC and CD points are mostly in the top right and left corners so agent 2 responds to cooperation with cooperation. Agents also cooperate at the start state, represented by $\varnothing$ points all hidden in the top right corner. Tit-for-tat strategy is further indicated by the losses close to 1 in part (B). On the other hand, most points for LA/CO/SGA/NL are accumulated at the bottom left, so agents mostly defect. This results in poor losses, demonstrating the limited effectiveness of recent proposals like SGA and CO. Finally note that trained parameters and losses for SOS are almost identical to those for LOLA, displaying equal capacity in opponent shaping while also inheriting convergence guarantees and outperforming LOLA in the next experiment.

**Tandem:** Results are given in Figure 3. SOS always succeeds in decreasing $p$ to reach the correct equilibria, with losses averaging at 0. LOLA fails to preserve fixed points, overshooting with losses averaging at $4/9$. The criterion for SOS is shown in action in part (B), decaying $p$ to avoid overshooting. This illustrates that purely *theoretical* guarantees descend into *practical* outperfor-

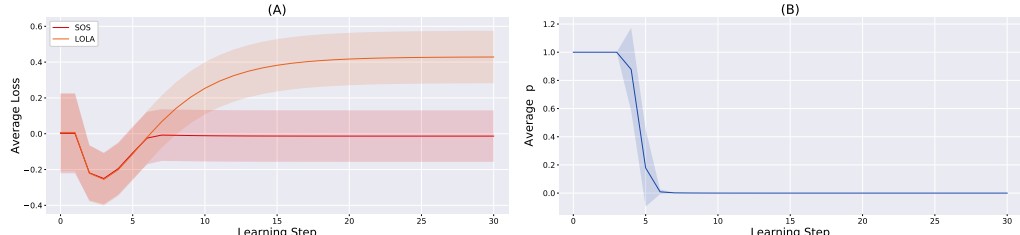

Figure 3: Results in the tandem game. **(A)** Average loss and **(B)** average $p$ at each learning step, across 300 runs, with shaded deviations. SOS decays $p$ to avoid arrogance and outperforms LOLA.

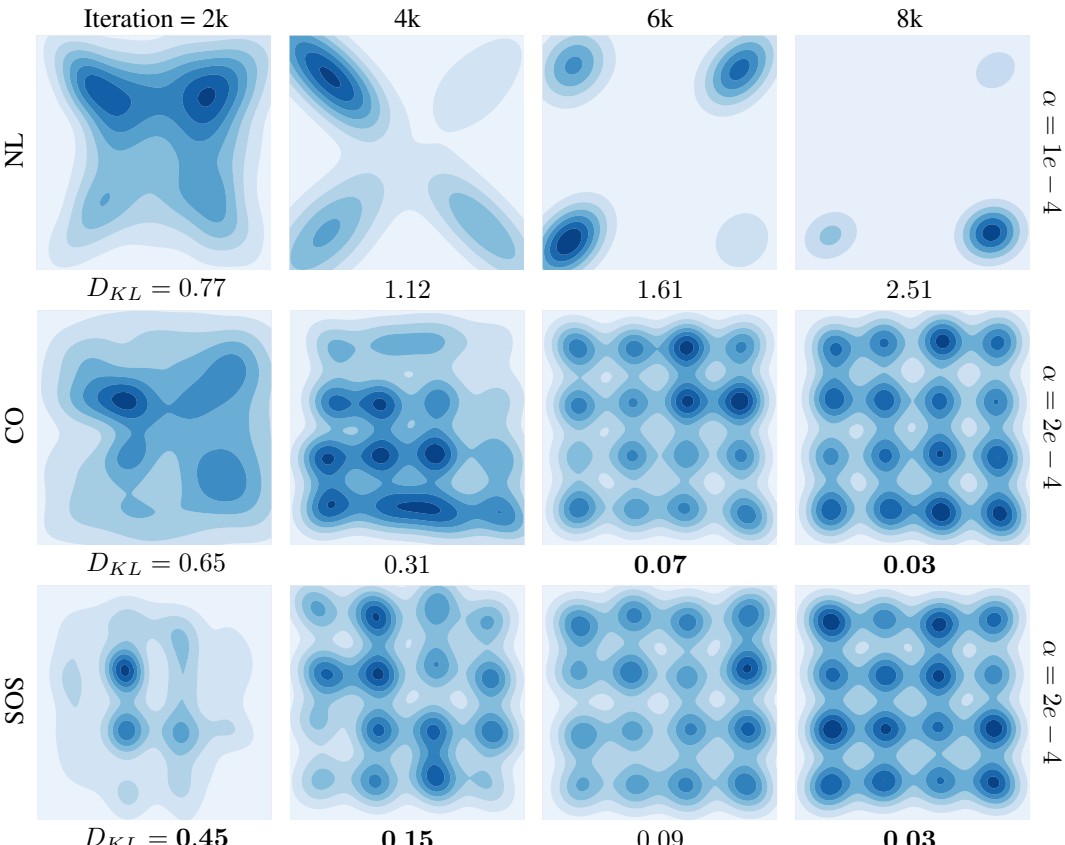

Figure 4: Generator distribution at sampled iterations. NL suffers from mode collapse and hopping, while CO and SOS learn the correct mixture of Gaussians. Below each plot: KL divergence $D_{KL}(P \parallel Q)$ from generator $P$ to ground truth $Q$, estimated from 25600 samples. To the RHS of each row: learning rate $\alpha$. Best result at each iteration shown in bold.

mance. Note that SOS even gets away from the LOLA fixed points if initialised there (not shown), converging to improved losses using the alignment criterion with LookAhead.

**Gaussian mixtures:** The generator distribution and KL divergence are given at $\{2k, 4k, 6k, 8k\}$ iterations for NL, CO and SOS in Figure 4. Results for SGA, LOLA and LA are in Appendix F. SOS achieves convincing results by spreading mass across all Gaussians, as do CO/SGA/LOLA. LookAhead is significantly slower, while NL fails through mode collapse and hopping. Only visual inspection was used for comparison by Balduzzi et al. (2018), while KL divergence gives stronger numerical evidence here. SOS and CO are slightly superior to others with reference to this metric. However CO is aimed specifically toward two-player zero-sum GAN optimisation, while SOS is widely applicable with strong theoretical guarantees in all differentiable games.

# 6 CONCLUSION

Theoretical results in machine learning have significantly helped understand the causes of success and failure in applications, from optimisation to architecture. While gradient descent on single losses has been studied extensively, algorithms dealing with interacting goals are proliferating, with little grasp of the underlying dynamics. The analysis behind CO and SGA has been helpful in this respect, though lacking either in generality or convergence guarantees. The first contribution of this paper is to provide a unified framework and fill this theoretical gap with robust convergence results for LookAhead in all differentiable games. Capturing stable fixed points as the correct solution concept was essential for these techniques to apply.

Furthermore, we showed that opponent shaping is both a powerful approach leading to experimental success and cooperative behaviour – while at the same time preventing LOLA from preserving fixed points in general. This conundrum is solved through a robust interpolation between LookAhead and LOLA, giving birth to SOS through a robust criterion. This was partially enabled by choosing to preserve the 'middle' term in LOLA, and using it to inherit stability from LookAhead. This results in convergence guarantees stronger than all previous algorithms, but also in practical superiority over LOLA in the tandem game. Moreover, SOS fully preserves opponent shaping and outperforms SGA, CO, LA and NL in the IPD by encouraging tit-for-tat policy instead of defecting. Finally, SOS convincingly learns Gaussian mixtures on par with the dedicated CO algorithm.

# 7 ACKNOWLEDGEMENTS

This project has received funding from the European Research Council under the European Union's Horizon 2020 research and innovation programme (grant agreement number 637713). It was also supported by the Oxford-Google DeepMind Graduate Scholarship.

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

# APPENDIX

## A    STABLE FIXED POINTS

In the main text we showed that Nash equilibria are inadequate in multi-agent learning, exemplified by the simple game given by $L^1 = L^2 = xy$, where the origin is a global Nash equilibrium but a saddle point of the losses. It is not however obvious that SFP are a better solution concept. We begin by pointing out that for single losses, invertibility and symmetry of the Hessian imply positive *definiteness* at SFP. These are exactly local minima of $L$ detected by the second partial derivative test, namely those points provably attainable by gradient descent.

To emphasise this, note that gradient descent does *not* converge locally to *all* local minima. This can be seen by considering the example $L(x, y) = y^2$ and the local (global) minimum $(0, 0)$. There is no neighbourhood for which gradient descent converges to $(0, 0)$, since initialising at $(x_0, y_0)$ will always converge to $(x_0, 0)$ for appropriate learning rates, with $x_0 \neq 0$ almost surely. This occurs precisely because the Hessian is singular at $(0, 0)$. Though a degenerate example, this suggests an important difference to make between the ideal solution concept (local minima) and that for which local convergence claims are possible to attain (local minima with invertible $H \succeq 0$).

Accordingly, the definition of SFP is the immediate generalisation of 'fixed points with positive semi-definite Hessian', or in other words, 'second-order-tractable local minima'. It is important to impose only positive *semi*-definiteness to keep the class as large as possible, despite strict positive definiteness holding for single losses due to symmetry. Imposing strict positivity would for instance exclude the origin in the cyclic game $L^1 = xy = -L^2$, a point certainly worthy of convergence.

Note also that imposing a weaker condition than $H \succeq 0$ would be incorrect. Invertibility aside, local convergence of gradient descent on single functions cannot be guaranteed if $H \nsucceq 0$, since such points are strict saddles. These are almost always avoided by gradient descent, as proven by Lee et al. (2016) and Panageas & Piliouras (2017). It is thus necessary to impose $H \succeq 0$ as a minimal requirement in optimisation methods attempting to generalise gradient descent.

**Remark A.1.** A matrix $H$ is positive semi-definite iff the same holds for its symmetric part $S = (H + H^\mathsf{T})/2$, so SFP could equivalently be defined as $S(\bar{\theta}) \succeq 0$. This is the original formulation given by part of the authors (Balduzzi et al., 2018), who also imposed the extra requirement $S(\theta) \succeq 0$ in a *neighbourhood* of $\bar{\theta}$. After discussion we decided to drop this assumption, pointing out that it is 1) more restrictive, 2) superficial to all theoretical results and 3) weakens the analogy with tractable local minima. The only thing gained by imposing semi-positivity in a neighbourhood is that SFP become a subset of Nash equilibria.

Regarding unstable fixed points and strict saddles, note that $H(\bar{\theta}) \succ 0$ implies $H(\theta) \succ 0$ in a neighbourhood, hence being equivalent to the definition in Balduzzi et al. (2018). It follows also that unstable points are a subset of strict saddles: if $H(\bar{\theta}) \prec 0$ then all eigenvalues are negative since any eigenpair $(v, \lambda)$ satisfies

$$0 > \mathrm{Re}(v^\mathsf{T} H v) = \mathrm{Re}(\lambda v^\mathsf{T} v) = \mathrm{Re}(\lambda).$$

We introduced strict saddles in this paper as a generalisation of unstable FP, which are more difficult to handle but nonetheless tractable using dynamical systems. The name is chosen by analogy to the definition in Lee et al. (2016) for single losses.

## B    LOLA VECTORIAL FORM

**Proposition B.1.** The LOLA gradient adjustment is

$$\mathrm{LOLA} = (I - \alpha H_o)\xi - \alpha \, \mathrm{diag}(H_o^\mathsf{T} \nabla L).$$

in the usual assumption of equal learning rates.

*Proof.* Recall the modified objective

$$L^1(\theta^1, \theta^2 - \alpha \nabla_2 L^2, \ldots, \theta^n - \alpha \nabla_n L^n)$$

for agent 1, and so on for each agent. First-order Taylor expansion yields

$$L^1 - \alpha \sum_{j \neq 1} (\nabla_j L^1)^\mathsf{T} \nabla_j L^j$$

and similarly for each agent. Differentiating with respect to $\theta^i$, the adjustment for player $i$ is

$$
\begin{aligned}
\text{LOLA}_i &= \nabla_i \left[ L^i - \alpha \sum_{j \neq i} (\nabla_j L^i)^\mathsf{T} \nabla_j L^j \right] \\
&= \nabla_i L^i - \alpha \sum_{j \neq i} (\nabla_{ji} L^i)^\mathsf{T} \nabla_j L^j + (\nabla_{ji} L^j)^\mathsf{T} \nabla_j L^i \\
&= \nabla_i L^i - \alpha \sum_{j \neq i} \nabla_{ij} L^i \nabla_j L^j - \alpha \sum_{j \neq i} (\nabla_{ji} L^j)^\mathsf{T} \nabla_j L^i \\
&= \xi_i - \alpha \sum_j (H_o)_{ij} \xi_j - \alpha \sum_j (H_o^\mathsf{T})_{ij} (\nabla L)_{ji} \\
&= \xi_i - \alpha (H_o \xi)_i - \alpha (H_o^\mathsf{T} \nabla L)_{ii} \\
&= \left[ \xi - \alpha H_o \xi - \alpha \operatorname{diag}(H_o^\mathsf{T} \nabla L) \right]_i
\end{aligned}
$$

and thus

$$\text{LOLA} = (I - \alpha H_o)\xi - \alpha \operatorname{diag}(H_o^\mathsf{T} \nabla L)$$

as required. $\qquad\square$

## C  TANDEM GAME

We provide a more detailed exposition of the tandem game in this section, including computation of fixed points for NL/LOLA and corresponding losses. Recall that the game is given by

$$L^1(x, y) = (x + y)^2 - 2x \qquad \text{and} \qquad L^2(x, y) = (x + y)^2 - 2y \,.$$

Intuitively, agents wants to have $x \approx -y$ since $(x + y)^2$ is the leading loss, but would also prefer to have positive $x$ and $y$. These are incompatible, so the agents must not be 'arrogant' and instead make concessions. The fixed points are given by

$$\xi = 2(x + y - 1) \begin{pmatrix} 1 \\ 1 \end{pmatrix} = 0 \,,$$

namely any pair $(x, 1 - x)$. The corresponding losses are $L^1 = 1 - 2x = -L^2$, summing to 0 for any $x$. We have

$$H = 2 \begin{pmatrix} 1 & 1 \\ 1 & 1 \end{pmatrix} \succeq 0$$

everywhere, so all fixed points are SFP. LOLA fails to preserve these, since

$$\chi = \operatorname{diag}(H_o^\mathsf{T} \nabla L) = 4 \operatorname{diag} \begin{pmatrix} 0 & 1 \\ 1 & 0 \end{pmatrix} \begin{pmatrix} x + y - 1 & x + y \\ x + y & x + y - 1 \end{pmatrix} = 4(x + y) \begin{pmatrix} 1 \\ 1 \end{pmatrix}$$

which is non-zero for any SFP $(x, 1 - x)$. Instead, LOLA can only converge to points such that

$$\text{LOLA} = \xi - \alpha H_o \xi - \alpha \chi = 0 \,.$$

We solve this explicitly as follows:

$$
\begin{aligned}
\text{LOLA} &= 2(x + y - 1) \begin{pmatrix} 1 \\ 1 \end{pmatrix} - 4\alpha(x + y - 1) \begin{pmatrix} 0 & 1 \\ 1 & 0 \end{pmatrix} \begin{pmatrix} 1 \\ 1 \end{pmatrix} - 4\alpha(x + y) \begin{pmatrix} 1 \\ 1 \end{pmatrix} \\
&= 2 \left[ (1 - 4\alpha)(x + y) - (1 - 2\alpha) \right] \begin{pmatrix} 1 \\ 1 \end{pmatrix} \,.
\end{aligned}
$$

The fixed points for LOLA are thus pairs $(x, y)$ such that

$$x + y = \frac{1 - 2\alpha}{1 - 4\alpha},$$

noting that $(1 - 2\alpha)/(1 - 4\alpha) > 1$ for all $\alpha > 0$. This leads to worse losses

$$L^1 = \left( \frac{1 - 2\alpha}{1 - 4\alpha} \right)^2 - 2x > 1 - 2x = L^1(x, 1 - x)$$

for agent 1 and similarly for agent 2. In particular, losses always sum to something greater than 0. This becomes negligible as the learning rate becomes smaller, but is always positive nonetheless Taking $\alpha$ arbitrarily small is not a viable solution since convergence will in turn be arbitrarily slow. LOLA is thus not a strong algorithm candidate for all differentiable games.

## D  CONVERGENCE PROOFS

We use Ostrowski's theorem as a unified framework for proving local convergence of gradient-based methods. This is a standard result on fixed-point iterations, adapted from (Ortega & Rheinboldt, 2000, 10.1.3). We also invoke and prove a topological result of our own, Lemma D.9, at the end of this section. This is useful in deducing local convergence, though not central to intuition.

**Theorem D.1** (Ostrowski). Let $F : \Omega \to \mathbb{R}^d$ be continuously differentiable on an open subset $\Omega \subseteq \mathbb{R}^d$, and assume $\bar{x} \in \Omega$ is a fixed point. If all eigenvalues of $\nabla F(\bar{x})$ are strictly in the unit circle of $\mathbb{C}$, then there is an open neighbourhood $U$ of $\bar{x}$ such that for all $x_0 \in U$, the sequence $F^{(k)}(x_0)$ converges to $\bar{x}$. Moreover, the rate of convergence is at least linear in $k$.

**Definition D.2.** A matrix $M$ is called *positive stable* if all its eigenvalues have positive real part.

Recall the simultaneous gradient $\xi$ and the Hessian $H$ defined for differentiable games. Let $X$ be any matrix with continuously differentiable entries.

**Corollary D.3.** Assume $\bar{x}$ is a fixed point of a differentiable game such that $XH(\bar{x})$ is positive stable. Then the iterative procedure

$$F(x) = x - \alpha X \xi(x)$$

converges locally to $\bar{x}$ for $\alpha > 0$ sufficiently small.

*Proof.* By definition of fixed points, $\xi(\bar{x}) = 0$ and so

$$\nabla[X\xi](\bar{x}) = \nabla X(\bar{x})\xi(\bar{x}) + X(\bar{x})\nabla \xi(\bar{x}) = XH(\bar{x})$$

is positive stable by assumption, namely has eigenvalues $a_k + ib_k$ with $a_k > 0$. It follows that

$$\nabla F(\bar{x}) = I - \alpha \nabla[X\xi](\bar{x})$$

has eigenvalues $1 - \alpha a_k - i\alpha b_k$, which are in the unit circle for small $\alpha$. More precisely,

$$|1 - \alpha a_k - i\alpha b_k|^2 < 1$$
$$\iff \quad 1 - 2\alpha a_k + \alpha^2 a_k^2 + \alpha^2 b_k^2 < 1$$
$$\iff \quad 0 < \alpha < \frac{2a_k}{a_k^2 + b_k^2}$$

which is always possible for $a_k > 0$. Hence $\nabla F(\bar{x})$ has eigenvalues in the unit circle for $0 < \alpha < \min_k 2a_k/(a_k^2 + b_k^2)$, and we are done by Ostrowski's Theorem since $\bar{x}$ is a fixed point of $F$. $\square$

We apply this corollary to LookAhead, which is given by

$$F(\theta) = \theta - \alpha X \xi(\theta)$$

where $X = (I - \alpha H_o)$. It is thus sufficient to prove the following result.

**Theorem D.4.** Let $H \succeq 0$ invertible with symmetric diagonal blocks. Then there exists $\epsilon > 0$ such that $(I - \alpha H_o)H$ is positive stable for all $0 < \alpha < \epsilon$.

**Remark D.5.** Note that $(I - \alpha H_o)H$ may fail to be positive *definite*, though true in the case of $2 \times 2$ matrices. This no longer holds in higher dimensions, exemplified by the Hessian

$$H = \begin{pmatrix} 9 & -4 & -3 & -3 \\ -2 & 1 & 2 & 1 \\ -3 & 0 & 1 & 0 \\ -3 & 1 & 2 & 1 \end{pmatrix}.$$

By direct computation (symbolic in $\alpha$), one can show that $G = (I - \alpha H_o)H$ always has positive eigenvalues for small $\alpha > 0$, whereas its symmetric part $S$ always has a negative eigenvalue with magnitude in the order of $\alpha$. This implies that $S$ and in turn $G$ is not positive definite. As such, an analytical proof of the theorem involving bounds on the corresponding bilinear form will fail.

This makes the result all the more interesting, but more involved. Central to the proof is a similarity transformation proving positive definiteness *with respect to a different inner product*, a novel technique we have not found in the multi-agent learning literature.

*Proof.* We cannot study the eigenvalues of $G$ directly, since there is no necessary relationship between eigenpairs of $H$ and $H_o$. In the aim of using analytical tools, the trick is to find a positive definite matrix which is similar to $G$, thus sharing the same positive eigenvalues. First define

$$G_1 = (I + \alpha H_d)H \qquad \text{and} \qquad G_2 = -\alpha H^2,$$

where $H_d$ is the sub-matrix of diagonal blocks,and rewrite

$$G = (I - \alpha H_o)H = (I - \alpha(H - H_d))H = (I + \alpha H_d)H - \alpha H^2 = G_1 + G_2.$$

Note that $H_d$ is block diagonal with symmetric blocks $\nabla_{ii}L^i \succeq 0$, so $(I + \alpha H_d)$ is symmetric and positive definite for all $\alpha \geq 0$. In particular its principal square root

$$M = (I + \alpha H_d)^{1/2}$$

is unique and invertible. Now note that

$$M^{-1}G_1M = M^{-1}M^2HM = M^\mathsf{T}HM,$$

which is positive semi-definite since

$$u^\mathsf{T}M^\mathsf{T}HMu = (Mu)^\mathsf{T}H(Mu) \geq 0$$

for all non-zero $u$. In particular $M$ provides a similarity transformation which eliminates $H_d$ from $G_1$ while simultaneously delivering positive semi-definiteness. We can now prove that

$$M^{-1}GM = M^{-1}G_1M + M^{-1}G_2M$$

is positive definite, establishing positive stability of $G$ by similarity. Let $m = d - 1$ where $d$ is the vector space dimension, namely $H \in \mathbb{R}^{d \times d}$. Recall that the $m$-sphere $S^m \subset \mathbb{R}^d$ is the space of unit vectors in $\mathbb{R}^d$. Take any $u \in S^m$ and consider the quantity

$$u^\mathsf{T}M^{-1}GMu.$$

First note that a Taylor expansion of $M$ in $\alpha$ yields

$$M = (I + \alpha H_d)^{1/2} = I + O(\alpha)$$

and

$$M^{-1} = (I + \alpha H_d)^{-1/2} = I + O(\alpha).$$

This implies in turn that

$$u^\mathsf{T}M^{-1}GMu = u^\mathsf{T}Gu + O(\alpha).$$

There are two cases to distinguish. If $u^\mathsf{T}Hu > 0$ then

$$u^\mathsf{T}M^{-1}GMu = u^\mathsf{T}Gu + O(\alpha) = u^\mathsf{T}G_1u + O(\alpha) = u^\mathsf{T}Hu + O(\alpha) > 0$$

for $\alpha$ sufficiently small. Otherwise, $u^\mathsf{T}Hu = 0$ and consider decomposing $H$ into symmetric and antisymmetric parts $S = (H + H^\mathsf{T})/2$ and $A = (H - H^\mathsf{T})/2$, so that $H = S + A$. By antisymmetry

of $A$ we have $u^\mathsf{T} Au = 0$ and hence $u^\mathsf{T} Hu = 0 = u^\mathsf{T} Su$. Now $H \succeq 0$ implies $S \succeq 0$, so by Cholesky decomposition of $S$ there exists a matrix $T$ such that $S = T^\mathsf{T} T$. In particular $0 = u^\mathsf{T} Su = \|Tu\|^2$ implies $Tu = 0$, and in turn $Su = 0$. Since $H$ is invertible and $u \neq 0$, we have $0 \neq Hu = Au$ and so $\|Au\|^2 > 0$. It follows in particular that

$$-\alpha u^\mathsf{T} H^2 u = -\alpha u^\mathsf{T} (S^\mathsf{T} - A^\mathsf{T})(S + A)u = \alpha u^\mathsf{T} A^\mathsf{T} Au = \alpha \|Au\|^2 > 0 \,.$$

Using positive semi-definiteness of $M^{-1} G_1 M$,

$$\begin{aligned}
u^\mathsf{T} M^{-1} GMu &= u^\mathsf{T} M^{-1} G_1 Mu + u^\mathsf{T} M^{-1} G_2 Mu \\
&\geq -\alpha u^\mathsf{T} M^{-1} H^2 Mu \\
&= -\alpha u^\mathsf{T} H^2 u + O(\alpha^2) \\
&= \alpha \|Au\|^2 + O(\alpha^2) > 0
\end{aligned}$$

for $\alpha > 0$ small enough. We conclude that for any $u \in S^m$ there is $\epsilon_u > 0$ such that

$$u^\mathsf{T} M^{-1} GMu > 0$$

for all $0 < \alpha < \epsilon_u$, where $g(\alpha, u) = u^\mathsf{T} M^{-1} GMu$ is a function $g : \mathbb{R}^+ \times S^m \to \mathbb{R}$ with $S^m$ compact. By Lemma D.9, this can be extended uniformly with some $\epsilon > 0$ such that

$$u^\mathsf{T} M^{-1} GMu > 0$$

for all $u \in S^m$ and $0 < \alpha < \epsilon$. It follows that $M^{-1} GM$ is positive definite for all $0 < \alpha < \epsilon$ and thus $G$ is positive stable for $\alpha$ in the same range, by similarity. $\qquad\square$

**Corollary D.6.** LookAhead converges locally to stable fixed points for $\alpha > 0$ sufficiently small.

*Proof.* For any SFP $\bar{\theta}$ we have $\xi(\bar{\theta}) = 0$ and $H(\bar{\theta}) \succeq 0$ invertible by definition, with diagonal blocks $\nabla_{ii} L^i$ symmetric by twice continuous differentiability. We are done by the result above and Corollary D.3. $\qquad\square$

We now prove that local convergence results descend to SOS. The following lemma establishes the crucial claim that our criterion for $p$ is $C^1$ in neighbourhoods of fixed points. This is necessary to invoke analytical arguments including Ostrowski's Theorem, and would be untrue globally.

**Lemma D.7.** If $\bar{\theta}$ is a fixed point and $\alpha$ is sufficiently small then $p = \|\xi\|^2$ in a neighbourhood of $\bar{\theta}$.

*Proof.* First note that $\xi(\bar{\theta}) = 0$, so there is a (bounded) neighbourhood $V$ of $\bar{\theta}$ such that $\|\xi(\theta)\| < b$ for all $\theta \in V$, for any choice of hyperparameter $b \in (0, 1)$. In particular $p_2(\theta) = \|\xi(\theta)\|^2$ by definition of the second criterion. We want to show that $p(\theta) = p_2(\theta)$ near $\bar{\theta}$, or equivalently $p_1(\theta) \geq p_2(\theta)$. Since $p_2(\theta) = \|\xi(\theta)\|^2 < b^2 < 1$ in $V$, it remains only to show that

$$\frac{-a\|\xi_0\|^2}{\langle -\alpha\chi, \xi_0 \rangle} \geq \|\xi(\theta)\|^2$$

in some neighbourhood $U \subseteq V$ of $\bar{\theta}$, for any choice of hyperparameter $a \in (0, 1)$. Now by boundedness of $V$ and continuity of $\chi$, there exists $c > 0$ such that $\|-\alpha\chi(\theta)\| = \alpha^2\|\chi(\theta)\| < c$ for all $\theta \in V$ and bounded $\alpha$. It follows by Cauchy-Schwartz that

$$\frac{-a\|\xi_0\|^2}{\langle -\alpha\chi, \xi_0 \rangle} \geq \frac{a\|\xi_0\|}{\|-\alpha\chi\|} > a\|\xi_0\|/c$$

in $V$. Now note that

$$\|\xi_0\| = \|(I - \alpha H_o)\xi\| \geq d\|\xi\|$$

in $V$, for some $d > 0$ and $\alpha$ sufficiently small, by boundedness of $V$ and continuity of $H_o$. Finally there is a sub-neighbourhood $U \subset V$ such that $\|\xi(\theta)\| < ad/c$ for all $\theta \in U$, so that $ad\|\xi\|/c > \|\xi(\theta)\|^2$ and hence

$$\frac{-a\|\xi_0\|^2}{\langle -\alpha\chi, \xi_0 \rangle} > \|\xi\|^2 = p_2$$

in $U$. Hence $p(\theta) = \min\{p_1(\theta), p_2(\theta)\} = p_2(\theta) = \|\xi(\theta)\|^2$ for all $\theta \in U$, as required. $\qquad\square$

**Theorem D.8.** SOS converges locally to stable fixed points for $\alpha > 0$ sufficiently small.

*Proof.* Though the criterion for $p$ is dual, we will only use the second part. More precisely,

$$p = \min\{p_1, p_2\} \leq p_2 = \|\xi\|$$

if $\|\xi\| < b$. The aim is to show that if $\bar{\theta}$ is an SFP then $\nabla \xi_p(\bar{\theta})$ is positive stable for small $\alpha$, using Ostrowski to conclude as usual. The first problem we face is that $\nabla \xi_p$ does not exist everywhere, since $p(\theta)$ is not a continuous function. However we know by Lemma D.7 that $p = \|\xi\|^2$ in a neighbourhood $U$ of $\bar{\theta}$, so $\xi_p$ is continuously differentiable in $U$. Moreover, $p(\bar{\theta}) = \|\xi(\bar{\theta})\|^2 = 0$ with gradient

$$\nabla p(\bar{\theta}) = 2H^\mathsf{T}\xi(\bar{\theta}) = 0$$

by definition of fixed points. It follows that

$$\nabla \xi_p(\bar{\theta}) = (I - \alpha H_o)H(\bar{\theta}) - \alpha \nabla p(\bar{\theta})\chi(\bar{\theta}) - \alpha p(\bar{\theta})\nabla\chi(\bar{\theta}) = (I - \alpha H_o)H(\bar{\theta})$$

which is identical to LookAhead. This is positive stable for all $0 < \alpha < \epsilon$, and $\bar{\theta}$ is a fixed point of the iteration since

$$\xi_p(\bar{\theta}) = (I - \alpha H_o)\xi(\bar{\theta}) - \alpha p(\bar{\theta})\chi(\bar{\theta}) = 0\,.$$

We conclude by Corollary D.3 that SOS converges locally to SFP for any $a, b \in (0, 1)$ and $\alpha$ sufficiently small. $\qquad\square$

**Lemma D.9.** Let $g : \mathbb{R}^+ \times Y \to Z$ continuous with $Y$ compact and $Z \subseteq \mathbb{R}$. Assume that for any $u \in Y$ there is $\epsilon_u > 0$ such that $g(\alpha, u) > 0$ for all $0 < \alpha < \epsilon_u$. Then there exists $\epsilon > 0$ such that $g(\alpha, u) > 0$ for all $0 < \alpha < \epsilon$ and $u \in Y$.

*Proof.* For any $u \in Y$ there is $\epsilon_u > 0$ such that

$$(0, \epsilon_u) \times \{u\} \subseteq g^{-1}(0, \infty)\,.$$

We would like to extend this uniformly in $u$, namely prove that

$$(0, \epsilon) \times Y \subseteq g^{-1}(0, \infty)\,.$$

for some $\epsilon > 0$. Now $g^{-1}(0, \infty)$ is open by continuity of $g$, so each $(0, \epsilon_u) \times \{u\}$ has a neighbourhood $X_u$ contained in $g^{-1}(0, \infty)$. Open sets in a product topology are unions of open products, so

$$X_u = \bigcup_x U_x \times V_x\,.$$

In particular $(0, \epsilon_u) \subseteq \bigcup_x U_x$ and at least one $V_x$ contains $u$, so we can take the open neighbourhood to be

$$X_u = (0, \epsilon_u) \times V_u \subseteq g^{-1}(0, \infty)$$

for some neighbourhood $V_u$ of $u$. In particular $Y \subseteq \bigcup_{u \in Y} V_u$, and by compactness there is a finite cover $Y \subseteq \bigcup_{i=1}^k V_{u_i}$. Letting $\epsilon = \min\{\epsilon_i\}_{i=1}^k > 0$, we obtain the required inclusion

$$(0, \epsilon) \times Y \subseteq (0, \epsilon) \times \bigcup_{i=1}^k V_{u_i} = \bigcup_{i=1}^k (0, \epsilon) \times V_{u_i} \subseteq \bigcup_{i=1}^k (0, \epsilon_i) \times V_{u_i} \subseteq g^{-1}(0, \infty)\,. \qquad\square$$

# E  NON-CONVERGENCE PROOFS

**Lemma E.1.** Let $a_k$ and $b_k$ be sequences of real numbers, and define $c_k = \min\{a_k, b_k\}$. If

$$L = \lim_{k \to \infty} c_k \qquad \text{and} \qquad L' = \lim_{k \to \infty} a_k$$

both exist then $L \leq L'$.

*Proof.* Assume for contradiction that $L > L'$, then there exists $\delta > 0$ such that $L > L' + \delta$. By definition of limits, there exist $M, N \in \mathbb{N}$ such that

$$|c_k - L| < \delta/2$$

and

$$|a_{k'} - L'| < \delta/2$$

for all $k \geq M$, $k' \geq N$. Expanding the absolute value, this implies

$$L - \delta/2 < c_k < L + \delta/2 \qquad \text{and} \qquad L' - \delta/2 < a_k < L' + \delta/2$$

for all $k \geq \max\{M, N\}$. Now $c_k \leq a_k$ for all $k$, hence

$$L - \delta/2 < c_k \leq a_k < L' + \delta/2$$

which implies the contradiction

$$L < L' + \delta\,. \qquad \qquad \square$$

**Proposition E.2.** If SOS converges to $\bar{\theta}$ and $\alpha > 0$ is small then $\bar{\theta}$ is a fixed point of the game.

*Proof.* The iterative procedure is given by

$$\theta_{k+1} = F(\theta_k) = \theta_k - \alpha\xi_p(\theta_k)\,.$$

If $\theta_k \to \bar{\theta}$ as $k \to \infty$ then taking limits on both sides of the iteration yields

$$\bar{\theta} = \bar{\theta} - \alpha \lim_{k\to\infty} \xi_p(\theta_k)$$

and so $\lim_k \xi_p(\theta_k) = 0$, omitting $k \to \infty$ for convenience. It follows by continuity that

$$\xi_0(\bar{\theta}) + \lim_k p(\theta_k) - \alpha\chi(\bar{\theta}) = 0\,,$$

noting that $p(\theta)$ is not a *globally* continuous function. Assume for contradiction that $\xi_0(\bar{\theta}) \neq 0$. There are two cases to distinguish for clarity.

(i) First assume $\langle -\alpha\chi, \xi_0\rangle(\bar{\theta}) \geq 0$. Note that $\lim_k p(\theta_k) \geq 0$ since $p(\theta) \geq 0$ for all $\theta$, and so

$$\langle\lim_k \xi_p(\theta_k), \xi_0(\bar{\theta})\rangle = \lim_k p(\theta_k)\langle -\alpha\chi, \xi_0\rangle(\bar{\theta}) + \|\xi_0(\bar{\theta})\|^2 > 0\,.$$

This is a contradiction since $\lim_k \xi_p(\theta_k) = 0$.

(ii) Otherwise, $\langle -\alpha\chi, \xi_0\rangle(\bar{\theta}) < 0$ and hence $\langle -\alpha\chi, \xi_0\rangle(\theta) < 0$ in a neighbourhood. In particular there exists $N \in \mathbb{N}$ such that

$$\langle -\alpha\chi, \xi_0\rangle(\theta_k) < 0$$

for all $k \geq N$. In particular

$$p_1(\theta_k) = \min\left\{1, \frac{-a\|\xi_0(\theta_k)\|^2}{\langle -\alpha\chi, \xi_0\rangle(\theta_k)}\right\}$$

for all $k \geq N$. Now notice that

$$\lim_k p(\theta_k) = \lim_k \min\left\{1, \frac{-a\|\xi_0(\theta_k)\|^2}{\langle -\alpha\chi, \xi_0\rangle(\theta_k)}, p_2(\theta_k)\right\}\,,$$

which implies

$$\lim_k p(\theta_k) \leq \lim_k \frac{-a\|\xi_0(\theta_k)\|^2}{\langle -\alpha\chi, \xi_0\rangle(\theta_k)} = \frac{-a\|\xi_0(\bar{\theta})\|^2}{\langle -\alpha\chi, \xi_0\rangle(\bar{\theta})}$$

by continuity and Lemma E.1. Finally we conclude

$$\langle\lim_k \xi_p, \xi_0\rangle(\theta_k) = \lim_k p(\theta_k)\langle -\alpha\chi, \xi_0\rangle(\bar{\theta}) + \|\xi_0(\bar{\theta})\|^2 \geq -a\|\xi_0(\bar{\theta})\|^2 + \|\xi_0(\bar{\theta})\|^2 > 0$$

for any $a \in (0, 1)$, a contradiction.

In both cases a contradiction is obtained, hence $\xi_0(\bar{\theta}) = 0 = (I - \alpha H_o)\xi(\bar{\theta})$. Now note that $(I - \alpha H_o)(\bar{\theta})$ is singular iff $H_o(\bar{\theta})$ has an eigenvalue $1/\alpha$, which is impossible for $\alpha$ sufficiently small. Hence $(I - \alpha H_o)\xi(\bar{\theta}) = 0$ implies $\xi(\bar{\theta}) = 0$, as required. $\qquad\square$

Now assume that $\theta$ is initialised randomly (or with arbitrarily small noise around a point), as is standard in ML. We prove that SOS locally avoids strict saddles using the Stable Manifold Theorem, inspired from Lee et al. (2017).

**Theorem E.3** (Stable Manifold Theorem). Let $\bar{x}$ be a fixed point for the $C^1$ local diffeomorphism $F : U \to \mathbb{R}^d$, where $U$ is a neighbourhood of $\bar{x}$ in $\mathbb{R}^d$. Let $E^s \oplus E^u$ be the generalised eigenspaces of $\nabla F(\bar{x})$ corresponding to eigenvalues with $|\lambda| \leq 1$ and $|\lambda| > 1$ respectively. Then there exists a *local stable center manifold $W$* with tangent space $E^s$ at $\bar{x}$ and a neighbourhood $B$ of $\bar{x}$ such that $F(W) \cap B \subset W$ and $\cap_{n=0}^{\infty} F^{-n}(B) \subset W$.

In particular, if $\nabla F(\bar{x})$ has at least one eigenvalue $|\lambda| > 1$ then $E^u$ has dimension at least 1. Since $W$ has tangent space $E^s$ at $\bar{x}$, with codimension at least one, it follows that $W$ has measure zero in $\mathbb{R}^d$. This is central in proving that the set of initial points in a neighbourhood which converge through SOS to a given strict saddle $\bar{\theta}$ has measure zero.

**Theorem E.4.** SOS locally avoids strict saddles almost surely, for $\alpha > 0$ sufficiently small.

*Proof.* Let $\bar{\theta}$ a strict saddle and recall that SOS is given by

$$F(\theta) = \theta - \alpha(I - \alpha H_o)\xi(\theta) + \alpha^2 p(\theta)\chi(\theta).$$

Recall by Lemma D.7 that $p(\theta) = \|\xi(\theta)\|^2$ for all $\theta$ in a neighbourhood $U$ of $\bar{\theta}$. Restricting $F$ to $U$, all terms involved are continuously differentiable and

$$\nabla F(\bar{\theta}) = I - \alpha(I - \alpha H_o)H(\bar{\theta})$$

by assumption that $\xi(\bar{\theta}) = 0$. Since all terms except $I$ are of order at least $\alpha$, $\nabla F(\bar{\theta})$ is invertible for all $\alpha$ sufficiently small. By the inverse function theorem, there exists a neighbourhood $V$ of $\bar{\theta}$ such that $F$ is has a continuously differentiable inverse on $V$. Hence $F$ restricted to $U \cap V$ is a $C^1$ diffeomorphism with fixed point $\bar{\theta}$.

By definition of strict saddles, $H(\bar{\theta})$ has a negative eigenvalue. It follows by continuity that $(I - \alpha H_o)H(\bar{\theta})$ also has a negative eigenvalue $a + ib$ with $a < 0$ for $\alpha$ sufficiently small. Finally,

$$\nabla F(\bar{\theta}) = I - \alpha(I - \alpha H_o)H(\bar{\theta})$$

has an eigenvalue $\lambda = 1 - \alpha a - i\alpha b$ with

$$|\lambda| = 1 - 2\alpha a + \alpha^2(a^2 + b^2) \geq 1 - 2\alpha a > 1.$$

It follows that $E^s$ has codimension at least one, implying in turn that the local stable set $W$ has measure zero. We can now prove that

$$Z = \{\theta \in U \cap V \mid \lim_{n \to \infty} F^n(\theta) = \bar{\theta}\}$$

has measure zero, or in other words, that local convergence to $\bar{\theta}$ occurs with zero probability. Let $B$ the neighbourhood guaranteed by the Stable Manifold Theorem, and take any $\theta \in Z$. By definition of convergence there exists $N \in \mathbb{N}$ such that $F^{N+n}(\theta) \in B$ for all $n \geq 0$, so that

$$F^N(\theta) \in \cap_{n=0}^{\infty} F^{-n}(B) \subset W$$

by the Stable Manifold Theorem. This implies that $\theta \in F^{-N}(W)$, and finally $\theta \in \cup_{n \in \mathbb{N}} F^{-n}(W)$. Since $\theta$ was arbitrary, we obtain the inclusion

$$Z \subseteq \cup_{n \in \mathbb{N}} F^{-n}(W).$$

Now $F^{-1}$ is $C^1$, hence locally Lipschitz and thus preserves sets of measure zero, so that $F^{-n}(W)$ has measure zero for each $n$. Countable unions of measure zero sets are still measure zero, so we conclude that $Z$ also has measure zero. In other words, SOS converges to $\bar{\theta}$ with zero probability upon random initialisation of $\theta$ in $U$. $\qquad\square$



Figure 5: Ground truth in the Gaussian mixture experiment.

## F  FURTHER GAUSSIAN MIXTURE EXPERIMENTS

In the Gaussian mixture experiment, data is sampled from a highly multimodal distribution designed to probe the tendency to collapse onto a subset of modes during training, given in Figure 5.

The generator distribution and KL divergence are given at {2k, 4k, 6k, 8k} iterations for LA, LOLA and SGA in Figure 6. LOLA and SGA successfully spread mass across all Gaussians. LookAhead displays mode collapse and hopping in early stages, but begins to incorporate further mixtures near $8k$ iterations. We ran further iterations and discovered that LookAhead eventually spreads mass across all mixtures, though very slowly. Comparing with results for NL/CO/SOS in the main text, we see that CO/SOS/LOLA/SGA are equally successful in qualitative terms.

Note that SOS/CO are slightly superior with respect to KL divergence after 6-8k iterations, though LOLA is initially faster. This may be due only to random sampling. We also noticed experimentally that LOLA often moves *away* from the correct distribution after 8-10k iterations (not shown), while SOS stays stable in the long run. This may occur thanks to the two-part criterion encouraging convergence, while LOLA continually attempts to exploit opponent learning.

Finally we plot $\|\xi\|$ at all iterations up to $12k$ for SOS, LA and NL in Figure 7 (other algorithms are omitted for visibility). This gives further evidence of SOS converging quite rapidly to the correct distribution, while NL perpetually suffers from mode hopping and LA lags behind significantly.

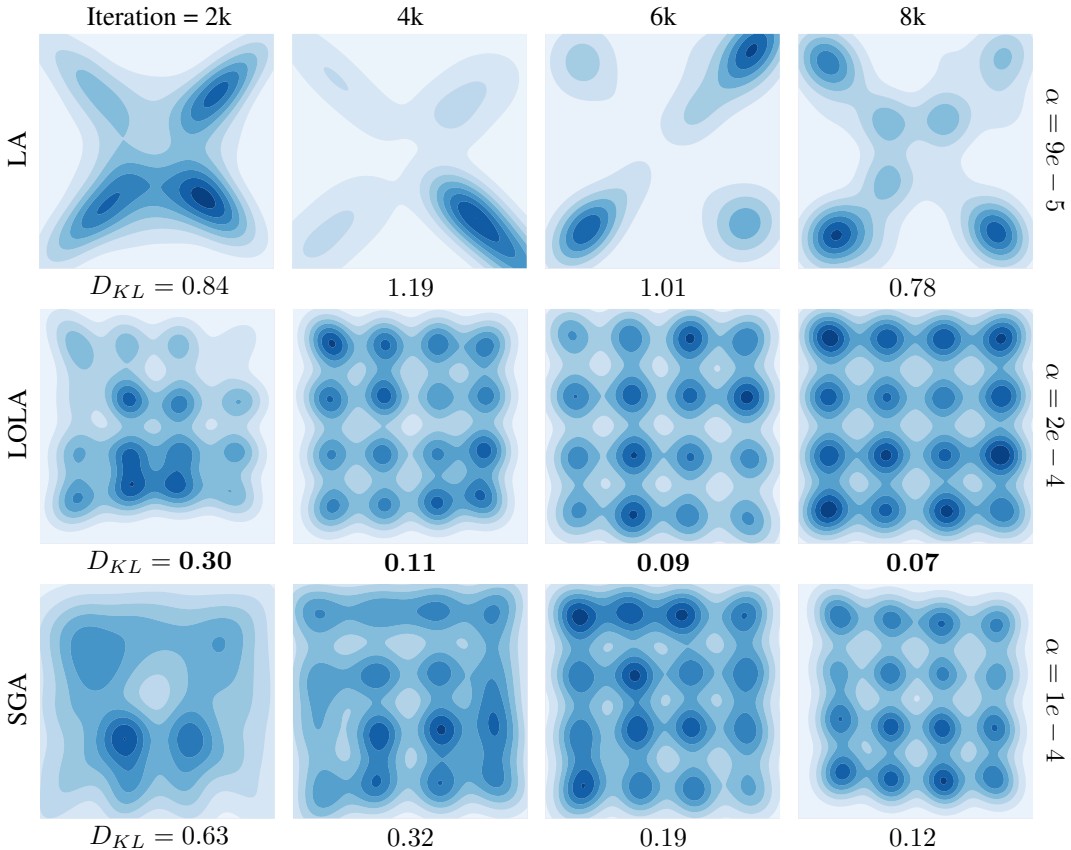

Figure 6: Generator distribution at sampled iterations for LA/LOLA/SGA. LA suffers in the early stages from mode collapse and hopping, but incorporates more mixtures later on. LOLA and SGA learn the correct mixture of Gaussians. Below each plot: KL divergence $D_{KL}(P \mid\mid Q)$ from generator $P$ to ground truth $Q$, estimated from 25600 samples. Best result at each iteration in bold.

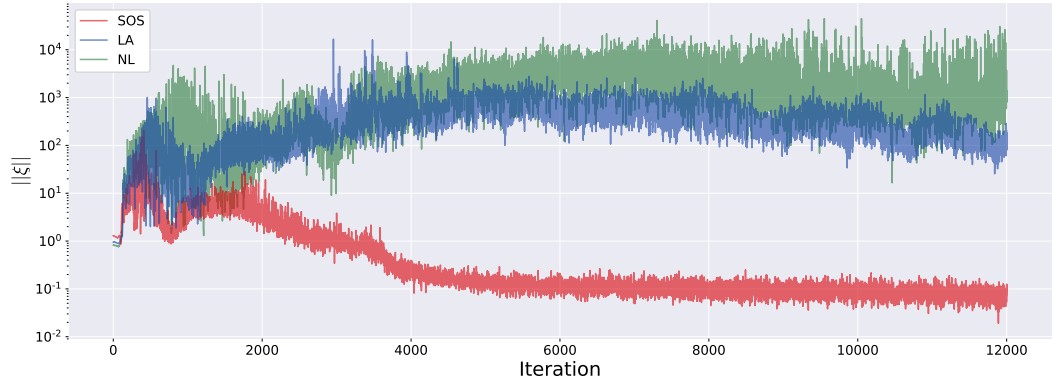

Figure 7: Semilog plot of $\|\xi\|$ at each iteration for SOS, LA and NL.

