# OpenReview forum: "Stable Opponent Shaping in Differentiable Games"
_ICLR.cc/2019/Conference_

### Official Review · AnonReviewer3 · 2018-11-02
**Stable Opponent Shaping in Differentiable Games**

**Rating:** 6
**Confidence:** 1

**Review:**

This paper introduces a new algorithm for differential game, where the goal is to find a optimize several objective functions simultaneously in a game of n players. The proposed algorithm is an interpolation between LOLA and LookAhead, and it perserves both the stability from LOLA and the "convergence to fixed point" property of LookAhead. The interpolation parameter is chosen in Section 3.2.

The paper looks novel, though some notations are not completely clear to me. For example, the defintions of the "current parameters" \hat{\theta}_1 and \hat{\theta}_2 in Section 3.1, and the stop-gradient operator. Also, how is the diag operator in Propostion 1 is defined? Normally it only represents the diagonal entries but here it might represent the diagonal blocks.

---

> ### Author Response · Authors · 2018-11-09
> **Response to your review**
>
> Thank you for this review. Some of the notation could certainly have been made clearer. Each point is addressed below and will be incorporated in a revision of the paper.
>
> 1. If agent $i$ has parameters $\theta^i_t$ at some fixed time $t$, the "current parameters" are simply defined as $\hat{\theta}^i = \theta^i_t$. The point is that these parameters are updated at each step to minimise a loss function. In LOLA, each agent assumes that the opponent updates their parameters dynamically, *after* their own optimisation step. In reality, they can only see the *current* parameters $\theta^i_t$ instead of the *optimised* (next) parameters $\theta^i_{t+1}$. Noticing this leads to an alternative algorithm, LookAhead.
>
> 2. The stop-gradient operator is really a *computational* operator rather than a formal, mathematical one. This is known in PyTorch as *detach* and in Tensorflow as *stop_gradient*. This operator acts on functions, setting their gradient to zero while keeping their value intact. In other words, $\bot f(x) = f(x)$ when evaluated at any $x$, while $\nabla (\bot f) (x) = 0$ for any $x$.
>
> 3. You are absolutely right: the diag operator in Proposition 1 should be defined as taking diagonal *blocks* since we are working with block matrices, not diagonal *entries*. Thank you for noticing this.

---

### Official Review · AnonReviewer2 · 2018-11-05
**Review for Stable Opponent Shaping in Differentiable Games**

**Rating:** 6
**Confidence:** 2

**Review:**

This paper studies differential games, in which there are n players and each has a loss function. The loss function depends on all parameters. Differential games appear naturally in GANs, where the two players are the generator and the discriminator. The authors first argue why Nash equilibria should not be the right solution concept for multi-agent learning and propose “stable fixed points” (SFP) as a possible solution concept. The authors then show the LOLA algorithm (Foerster et al. (2018)) fails to preserve fixed points by explicitly constructing an instance (the tandem game). In fact in the tandem game, LOLA will converge to sub-optimal scenarios with worse losses for both agents. The authors then show that an known algorithm LookAhead (Zhang & Lesser (2010)) has local convergence to SPF. However, LookAhead does not have the capacity to exploit opponent dynamics and encourage cooperation. To alleviate this issue, the authors propose a new algorithm SOS, which can be seen as an interpolation between LOLA and LookAhead, characterized by a parameter p. The authors also discuss how to choose the parameter p and prove that SOS will have local convergence to SFP and can avoid strict saddles.

Overall, this paper is well-written and develops algorithms for a well-motivated problem. Although I am not an expert on this topic, the paper seems interesting to me.

Minor Comment:
First paragraph in Section 2.2, "It is highly undesirable to converge to Nash in this game" -> Nash equilibria

---

> ### Author Response · Authors · 2018-11-09
> **Response to your review**
>
> Thank you for this review. We will be sure to incorporate your comment in a revision of the paper.

---

### Official Review · AnonReviewer1 · 2018-11-05
**Interesting paper, strong theoretical results but concerns with the main theorem**

**Rating:** 8
**Confidence:** 4

**Review:**

This paper focuses on the problem of convergence in multi-objective optimisation with differentiable losses. This topic is timely and relevant, given the increasing amount of recent work on multi-objective architectures, e.g. GANs, adversarial learning, multi-agent reinforcement learning. The authors focus on stable fixed points (SFP), rather than Nash equilibria, as the solution concept in the entirety of their analysis. Casting the recently proposed LOLA gradient adjustment into a general matrix form, they diagnose an example where the shaping term in LOLA prevents convergence to SFP. They also find that discarding the shaping term leads to an earlier method (which they name ''LA'') with convergence guarantees in two-player two-action games. However, this also loses the opponent shaping ability of LOLA. To address these limitation, the authors propose SOS, which interpolates between LA and LOLA, and dynamically chooses the interpolation coefficient $p$ so that their adjusted gradient preserves LOLA's shaping ability only to the extent allowed by the constraint of moving in LA's direction. The main goal of the paper is to show that SOS converges locally to SFP, and to fixed points only, while avoiding strict saddles. Experiments on synthetic games show that SOS preserves the benefit of LOLA while avoiding its theoretically-predicted issues, and a more complex Gaussian mixture GAN experiment shows SOS is empirically competitive with other gradient adjustment methods.

The main conceptual novelty consists of the dynamic interpolation term to combine advantages of LOLA and LA while avoiding pitfalls of both. The major strength of the paper lies in the clear justification for this interpolation approach. The paper contains strong theoretical results for general differentiable games, and deserves the notice of the ICLR community if valid. However, I have major concerns with the proof of Theorem 2 (i.e. Theorem D.4 in the appendix), which affects the validity of Corollary 3 and Theorem 4.

In the proof of Theorem D.4:
1. How does the expression $u^T M^{-1}GMu$ have conformable dimensions, when $G \in R^{d \times d}$ while $u \in R^{d-1}$? Was any assumption made about the matrix $M = (I + \alpha H_d)^{1/2}$?
2. In the middle of page 14, a unit vector $u \in S^m$ is defined, but it is not clear what vector space is meant by $S^m$.
3. In the second-to-last line of page 14, a quantity $S$ is used but not defined clearly in any preceding part of the proof. Remark D.5 refers to $S$ as the symmetric part of $G$, and asserts that S is not positive definite. If the quantity $S$ used in the proof is the same non-PD quantity, then $S$ does not have a Cholesky factorisation. So how is Cholesky decomposition conducted at end of page 14?
4. In the first line of page 15, a quantity $A$ is used but not defined anywhere else in the entire paper.
5. From the subsequent line, it appears to be the anti-symmetric part of H. Is it correct assumption? If so, $H^2$ is not $(S^T - A^T)(S + A)$. If you replace it with correct form, whole quantity does not compute to be positive or becomes meaningless.

As Theorem 2 is the crux for all the theoretical advancement presented in the paper, clarifications on above correctness questions is very important for clear acceptance of this work.

While Definition 1 precisely defines differentiable games to have *twice* differentiable losses, why do the authors assume *thrice* differentiable losses at the start of Section 4?

In Section 2.2, the authors make a broad statement that ''Nash equilibria cannot be the right solution concept for multi-agent learning.'' They provide one example where Nash is undesirable (L^1 = L^2 = xy). However, since this example can be viewed as a fully cooperative game with joint loss L = 2xy, it does not support the broader statement that Nash is undesirable in all games. Because this statement directly motivates the authors to focus on stable fixed points, rather than Nash, as the solution concept in their subsequent analysis, it is very important to provide better justification for the claim.

Minor comments:
1. Under Proposition 1, the authors suddenly speak of ''...the policy being optimal''. Since the author's work pertains to general multi-objective settings, not solely multi-agent reinforcement learning, the word ''policy'' sounds strange in context.
2. The statement of Proposition B.1, and the concluding line of the derivation, left out a coefficient $\alpha$ that is present in Proposition 1 in the main text.
3. While the authors claim and prove independence of theoretical results from choice of a and b, are there any practical implications in terms of performance or convergence?

---

> ### Author Response · Authors · 2018-11-09
> **Response to your review**
>
> Thank you for your detailed and thoughtful comments. Below we address each point regarding the proof of Theorem D.4. We will also revise the paper to clarify these points and want to emphasise that these details do not affect the validity of our results.
>
> 1. This is a notational confusion: $u$ lives in $R^d$, not $R^{d-1}$, while $G$ and $M$ are both square $d \times d$ matrices. Indeed $u$ is defined to be an arbitrary vector in $S^{d-1}$, the unit (d-1)-sphere living in Euclidian space $R^d$. This is a standard but confusing convention (see https://en.wikipedia.org/wiki/N-sphere ).
>
> 2. As above, $S^m$ with $m = d-1$ is the space of unit vectors in $R^d$.
>
> 3. $S$ and $A$ are the symmetric and antisymmetric parts of $H$ respectively, which we mistakenly failed to define in the paper. The definitions are $S = (H+H^T)/2$ and $A = (H-H^T)/2$, so that $H = S + A$. In the specific example of Remark D.5, $S$ is not positive definite. However, one can easily show that a matrix $H$ is positive semi-definite iff its symmetric part $S$ is positive semi-definite (consider $u^T H u = u^T S u + u^T A u = u^T S u$ by antisymmetry of $A$). By assumption in Theorem D.4, it follows that $S$ is positive semi-definite and thus has a Cholesky decomposition.
>
> 4. See point 3.
>
> 5. This is the correct assumption. Regarding your concern about the expression for $H^2$, we have $H = S + A$ but also $H = S^T - A^T$ by symmetry of $S$ and antisymmetry of $A$. It follows that $H^2 = (S^T - A^T)(S + A)$ as claimed.
>
> Definition 1 was chosen to be in line with prior work (Balduzzi et al, ICML 2018), where losses are *twice* differentiable. Our results require *thrice* differentiable losses because both Ostrowski and Stable Manifold Theorems require continuous differentiability of the gradient adjustment. Now the gradient adjustment for SOS contains second-order gradients of the losses through the Hessian $H$, so will only be continuously differentiable if the losses themselves are *thrice* continuously differentiable. We chose to make this extra (very weak) assumption explicit before stating our results, instead of changing the definition of differentiable games to fit our purposes. We are happy to alter the definition if this helps at all.
>
> Appendix A provides a more detailed justification for choosing stable fixed points over Nash equilibria as the correct solution concept for gradient-based optimisation in games. Though the example given in the main body is insufficient by itself, the aim was not to show that Nash are *always* undesirable (this is not true), but to show that optimisation algorithms should not aim/succeed in converging to *all* Nash equilibria. The appendix was referenced for further detail about stable fixed points, but we will further clarify this in the main paper in the final version.
>
> Replies to minor comments:
>
> 1. Agreed: speaking of "policy" is indeed too specific and inappropriate.
>
> 2. Well-spotted typo! We will correct this.
>
> 3. Choosing $a$ closer to $0$ means that SOS is forced to agree strongly with the direction of LA, while $a$ close to $1$ gives more flexibility (larger angle between the adjustments). In other words: smaller $a$ means potentially faster convergence, larger $a$ allows for more opponent shaping. Similarly for $b$: the parameter $p$ will be shrunk in a $b$-neighbourhood of fixed points, so larger $b$ ensures convergence in a wider radius while smaller $b$ allows for more opponent shaping. As briefly mentioned in the paper, we found that these hyperparameters were quite robust in experiments overall, though choosing $b = 0.1$ (quite small) for the IPD and Gaussian Mixtures was necessary to guarantee strong opponent shaping in a large region of parameter space. We hope this helps shed some light on the practical implications of choice on $a$ and $b$, though all theoretical results are indeed independent from this choice.

---

> > ### Comment · AnonReviewer1 · 2018-11-26
> > **Thanks for the response**
> >
> > Thank you for addressing all the comments.
> >
> > - I am satisfied with the explanation from the authors regarding Theorem D.4 and the revision adequately addresses most of the comments.
> >
> > - Regarding differentiability, it is fine to retain the original definition of differentiable games while your result requiring thrice differentiable losses. However, the justification you provided in your response needs to be added in to the paper and preferably as a note immediately after Definition 1 to avoid the misinterpretation.
> >
> > - Another question Lemma D.7 - Towards the end of proof, the application of Cauchy Schwartz is not clear. You show that ||-\alphaX(\theta)|| = \alpha^2||X(\theta)|| < c. However, it is not clear how the equation below that holds? Specifically, why does the negative sign in the first fraction disappear and somehow the overall term becomes >= \alpha||\Psi_0||/||-\alphaX||.
> >
> > It is recommended that the authors proofread all the proofs and equations and try to use notations and show derivations without making them confusing for the overall presentation. It would also help to number equations for quick reference.
> >
> > Overall the paper presents strong theoretical results with adequate empirical evidence. It certainly addresses an important problem of trade-off between convergence and stability in multi-objective settings and I have updated my score from 6 to 8 to strongly support it for acceptance.

---

> > > ### Author Response · Authors · 2018-11-27
> > > **Response**
> > >
> > > Thank you for reading through our clarifications in detail, and for providing further comments.
> > >
> > > - We will be sure to clarify the question of twice/thrice differentiability in a note following Definition 1.
> > >
> > > - The application of Cauchy-Schwartz goes as follows. Writing $u = -\alpha \chi$ and $v = \xi_0$, we have $-||u|| * ||v||  \leq <u, v>$ by (one half of) the Cauchy-Schwartz inequality. Taking opposites and inverses on both sides, we obtain $1/(||u|| * ||v||) \leq -1/<u, v>$. This is how the negative sign disappears from one fraction to the next. We will add an extra step in the equation to make this clearer.
> > >
> > > We will add derivations and check our proofs/equations before submitting a final revision. Thanks again for your helpful and thorough review.

---

### Public Comment · (anonymous) · 2018-11-26
**Related literature on stable equilibria and continuous games**

I was disappointed to see that the authors make no reference to the rich literature on continuous games where the positive-definiteness of the Hessian has been explored quite extensively as a stability criterion.

The role of this condition dates back (at least) to the work of Rosen in the 60's (Econometrica, 1965), wherein it was introduced precisely as a stability criterion for the convergence of first-order learning methods in N-player games with continuous action sets.

For a more recent take, the authors might also want to consult the monograph of Facchinei and Kanzow (Annals of OR, 2010): Hessian stability is discussed extensively in Section 5 of said paper (Algorithms), and plays the same role as in the current paper.

It should be noted that the above papers concern a model which is (in at least one sense) even more general than that of the authors, because the admissible actions of a player may depend on the actions of all other players (hence the term "generalized Nash equilibrium problem"/GNEP). Also, even though the above papers concern games with individually convex loss functions, the extension to non-convex games under local stability conditions has also been explored in the literature - see e.g., the recent preprint https://arxiv.org/abs/1608.07310.

The above goes to show that statements like "the only theoretical work on general game dynamics is Symplectic Gradient Adjustment (SGA) by Balduzzi et al. (2018)" are not representative of the state of the art in the subject. The same also holds for the authors' complete lack of references to this literature in Section 2.2 (and, to be clear, the papers mentioned above comprise but a small sample of a very active literature on games with continuous action spaces).

To state things frankly, the field is not a virgin territory only recently discovered, so I would urge the authors to take this into account in their bibliographical policy - the papers above could provide a starting point in that respect, so they should be properly cited and discussed.

---

> ### Author Response · Authors · 2018-11-28
> **Thank you for these references**
>
> Thank you for these important references. Unfortunately we were not aware of this literature, especially the monograph of Facchinei and Kanzow and the older work mentioned. Thanks also for linking to the preprint on general games with continuous action sets. This is a great starting point to explore further in this area: apologies for having been unaware of this in the first place. We will be sure to cite and discuss a number of these related works in a revision of the paper.

---

### Meta-Review · Area_Chair1 · 2018-12-14
**Correct and reasonably well-written paper with some concerns on missing literature**

**Confidence:** 3
**Recommendation:** Accept (Poster)

**Metareview:**

This paper provides interesting results on convergence and stability in general differentiable games. The theory appears to be correct, and the paper reasonably well written. The main concern is in connections to an area of related work that has been omitted, with overly strong statements in the paper that there has been little work for general game dynamics. This is a serious omission, since it calls into question some of the novelty of the results because they have not been adequately placed relative to this work. The authors should incorporate a thorough discussion on relations to this work, and adjust claims about novelty (and potentially even results) based on that literature.